# Improving Generalizability of PET DL Algorithms: List-Mode Reconstructions Improve DOTATATE PET Hepatic Lesion Detection Performance

**DOI:** 10.3390/bioengineering11030226

**Published:** 2024-02-27

**Authors:** Xinyi Yang, Michael Silosky, Jonathan Wehrend, Daniel V. Litwiller, Muthiah Nachiappan, Scott D. Metzler, Debashis Ghosh, Fuyong Xing, Bennett B. Chin

**Affiliations:** 1Department of Biostatistics and Informatics, University of Colorado Anschutz Medical Campus, Aurora, CO 80045, USA; xinyi.2.yang@cuanschutz.edu (X.Y.); debashis.ghosh@cuanschutz.edu (D.G.); fuyong.xing@cuanschutz.edu (F.X.); 2Department of Radiology, University of Colorado Anschutz Medical Campus, Aurora, CO 80045, USA; michael.silosky@cuanschutz.edu (M.S.); muthiah.nachiappan@cuanschutz.edu (M.N.); 3Department of Radiology, Santa Clara Valley Medical Center, San Jose, CA 95128, USA; 4GE HealthCare, Denver, CO 80222, USA; daniel.litwiller@ge.com; 5Department of Radiology, University of Pennsylvania, Philadelphia, PA 19104, USA; scott.metzler@uphs.upenn.edu; 6The Computational Bioscience Program, University of Colorado Anschutz Medical Campus, Aurora, CO 80045, USA; 7University of Colorado Cancer Center, University of Colorado Anschutz Medical Campus, Aurora, CO 80045, USA

**Keywords:** deep learning, convolutional neural network, gastroenteropancreatic neuroendocrine tumor, GEP-NET, DOTATATE, positron emission tomography, PET, liver tumor

## Abstract

Deep learning (DL) algorithms used for DOTATATE PET lesion detection typically require large, well-annotated training datasets. These are difficult to obtain due to low incidence of gastroenteropancreatic neuroendocrine tumors (GEP-NETs) and the high cost of manual annotation. Furthermore, networks trained and tested with data acquired from site specific PET/CT instrumentation, acquisition and processing protocols have reduced performance when tested with offsite data. This lack of generalizability requires even larger, more diverse training datasets. The objective of this study is to investigate the feasibility of improving DL algorithm performance by better matching the background noise in training datasets to higher noise, out-of-domain testing datasets. ^68^Ga-DOTATATE PET/CT datasets were obtained from two scanners: Scanner1, a state-of-the-art digital PET/CT (GE DMI PET/CT; n = 83 subjects), and Scanner2, an older-generation analog PET/CT (GE STE; n = 123 subjects). *Set1*, the data set from Scanner1, was reconstructed with standard clinical parameters (5 min; Q.Clear) and list-mode reconstructions (VPFXS 2, 3, 4, and 5-min). *Set2*, data from Scanner2 representing out-of-domain clinical scans, used standard iterative reconstruction (5 min; OSEM). A deep neural network was trained with each dataset: Network1 for Scanner1 and Network2 for Scanner2. DL performance (Network1) was tested with out-of-domain test data (*Set2*). To evaluate the effect of training sample size, we tested DL model performance using a fraction (25%, 50% and 75%) of *Set1* for training. Scanner1, list-mode 2-min reconstructed data demonstrated the most similar noise level compared that of *Set2*, resulting in the best performance (*F*_1_ = 0.713). This was not significantly different compared to the highest performance, upper-bound limit using in-domain training for Network2 (*F*_1_ = 0.755; *p*-value = 0.103). Regarding sample size, the *F*1 score significantly increased from 25% training data (*F*_1_ = 0.478) to 100% training data (*F*_1_ = 0.713; *p* < 0.001). List-mode data from modern PET scanners can be reconstructed to better match the noise properties of older scanners. Using existing data and their associated annotations dramatically reduces the cost and effort in generating these datasets and significantly improves the performance of existing DL algorithms. List-mode reconstructions can provide an efficient, low-cost method to improve DL algorithm generalizability.

## 1. Introduction

Gastroenteropancreatic neuroendocrine tumors (GEP-NETs) are most accurately imaged with ^68^Ga- and ^64^Cu-DOTATATE positron emission tomography/computed tomography (DOTATATE PET/CT), which are standard-of-care imaging modalities for tumor detection and staging [1,2,3]. Many computerized methods have been applied to automatic lesion detection and/or quantification in PET images [4], leading to improved objectivity and efficiency compared with manual tumor identification.

In recent years, the use of Deep Learning (DL) methods to identify and quantify lesions in PET/CT images has become a growing area of research. Deep neural networks have recently shown excellent performance [4,5] in quantifying uptake from a variety of radiopharmaceuticals including ^18^F-FDG [6,7,8], ^18^F-PSMA [9,10], ^68^Ga-PSMA [11] and ^68^Ga- and ^64^Cu-DOTATATE [12]. Previous studies have shown impressive results when the training data and the test data are from the same domain, which means they have the same or similar data distribution. However, training the networks usually requires a large amount of well-annotated data. In real-world clinical practice, it is difficult to collect and annotate enough data for model training because GEP-NETs are rare tumors, and lesion annotation in PET images is costly and challenging. When training a model based on a well-annotated dataset from a different site/scanner, the model usually typically shows degradation in performance when tested on a different unseen target dataset. This is because datasets from different sites/scanners usually exhibit different data distributions, i.e., domain shift. This domain shift is derived from several image parameters, such as differences in spatial resolution, image noise and image processing [13]. In PET images, it has been shown that background activity and noise have a significant impact on the detectability of lesions [14]. Collecting and annotating another large dataset with similar properties to the out-of-domain target dataset is extremely time and resource consuming, making this effectively unfeasible.

List-mode reconstructions allow existing data and their associated annotations to be retrospectively reconstructed with numerous variations to better match differences in other unseen dataset properties. Changing the reconstruction parameters can simulate different out-of-domain properties [15,16,17]. Thus, using existing datasets and their associated annotations with list-mode reconstructions may dramatically reduce the cost and effort to generate these better matching datasets. The purpose of this study is to investigate the feasibility of using list-mode reconstructions to better match image noise between training and out-of-domain testing datasets to improve the performance of lesion detection using deep neural networks in DOTATATE PET. In this study, we generated a set of list-mode reconstructed datasets with different acquisition times based on the same dataset, such that only the noise level is different between these reconstructed datasets. In addition, we also investigated the effect of training sample size on the cross-domain performance of deep neural networks. Based on the findings in this manuscript, reconstructing the existing PET data helps to significantly improve the performance of DL algorithms, in a low-cost and efficient manner. In this article, we introduce materials and methods in Section 2. The results are in Section 3, followed by our discussion in Section 4 and our conclusion in Section 5.

## 2. Materials and Methods

### 2.1. Image Acquisition and Datasets

This study was approved and performed under a waiver of informed consent from the Institutional Review Board at the University of Colorado Anschutz Medical Campus. All consecutive DOTATATE studies from our institution were de-identified using a three-digit numerical ID. The standard clinical acquisition and processing protocols were used as previously described [12]. Briefly, subjects with normal liver uptake, and those with 10 or fewer non-confluent hepatic lesions were included. Two separate ^68^Ga-DOTATATE PET image datasets from two different PET/CT scanners were included. The first dataset (*Set1*) comprised 83 subjects, of which 42 were normal and 41 were abnormal scans, with 134 hepatic lesions acquired from *Scanner1*, a modern digital PET/CT scanner (GE Discovery MI PET/CT, GE HealthCare, Waukesha, WI, USA). This scanner has time-of-flight (TOF) temporal resolution of approximately 380 ps. The second dataset (*Set2*) comprised 123 ^68^Ga-DOTATATE PET/CTs with 233 hepatic lesions acquired from *Scanner2*, an older generation photomultiplier tube-based PET/CT scanner (GE Discovery STE, GE HealthCare, Waukesha, WI, USA) [12]. *Set2* included 56 abnormal cases and 67 normal subjects. Following previous reports [11,12], we randomly split each dataset into 60%, 20% and 20% for training, validation and testing, respectively.

For *Set1*, images were reconstructed with the full 5 min of data using block sequential regularized expectation-maximization penalized-likelihood TOF reconstruction (BSREM, aka Q.Clear, GE HealthCare, Waukesha, WI, USA) with a Beta value of 400, a 256 × 256 matrix and a 70 cm reconstructed diameter resulting in voxels with dimensions of 2.73 mm × 2.73 mm × 2.79 mm. CT based attenuation correction was applied along with time-of-flight correction, point spread function recovery and scatter and decay corrections. Following the clinical reconstruction, list-mode data were utilized to generate additional TOF reconstructions with data-acquisition times of 2, 3, 4 and 5 min using conventional iterative reconstruction (TOFOSEM-PSF, aka VPFXS, GE HealthCare, Waukesha, WI, USA) with 3 iterations/16 subsets, a 192 × 192 matrix and a 70 cm reconstructed diameter resulting in voxels of 3.64 mm × 3.64 mm × 2.79 mm. Again, CT based attenuation correction was applied along with point spread function recovery and scatter and decay corrections. Additionally, these reconstructions were smoothed with a 5 mm Gaussian post-reconstruction filter. From *Set1*, we have 5 different sets of reconstructions: VPFXS 2 min, VPFXS 3 min, VPFXS 4 min, VPFXS 5 min and Q.Clear.

For *Set2*, PET images from clinical ^68^Ga DOTATATE PET/CT were also acquired with 5 min of acquisition time per bed position. These images were reconstructed using the full 5 min of data using ordered subset expectation maximization reconstruction (OSEM) with 3 iterations/16 subsets, a 128 × 128 matrix and a 60 cm reconstructed diameter resulting in voxels of 4.69 mm × 4.69 × 3.27 mm. Again, CT-based attenuation correction was applied along with scatter and decay corrections followed by a 5 mm Gaussian post-reconstruction filter. Point spread function recovery was not used as it was not available on this older scanner model.

### 2.2. Image Segmentation and Lesion Contours

Lesion segmentation was performed on all clinical reconstructions using a semiautomated MIM workflow (MIM version 7.03) as previously described [12]. This tool utilizes a modified PERCIST threshold based on regions of interest (ROI) placed in normal liver background which provides SUV*_mean_* and standard deviation of ^68^Ga DOTATATE activity. Lesions were identified using this threshold, defined as 1.5 times SUV*_mean_* plus 2 standard deviations of normal liver background. Once lesions were detected and visually confirmed, contours were refined using a commercially available gradient edge detection tool (PET Edge plus; MIM software 7.0.3). For the training and validation sets of reconstructions, contours generated using the high quality Q.Clear reconstruction were transferred to each of the VPFXS reconstructions.

### 2.3. Quantification of Image Noise

To quantify the difference in image noise, SUV measurements from the background ROIs from each reconstruction were used. The same number of subject samples (n = 25) was used to calculate the background noise characteristics from reconstructions of *Set1* and *Set2*. The SUV*_mean_* and standard deviation in normal liver background were recorded for each reconstruction and the coefficient of variation (COV) was calculated as the standard deviation of the ROI divided by SUV*_mean_*. The average and standard deviation of the COV across all subjects within each reconstruction type was also calculated. To determine if differences in COV between reconstructions were statistically significant, a series of paired *t*-tests was performed, comparing COV for each subject’s reconstructions between each combination of two reconstruction approaches.

### 2.4. Network Architecture

The lesion detection network in this study was built on a modified U-Net architecture [12], which has shown impressive performance for lesion identification in PET images (Appendix A). It consists of four residual learning blocks [18] in the downsampling path and the upsampling path, respectively. It also has two transposed convolutional layers [19] in the upsampling path for contextual information aggregation [20]. We optimized this network using a linear combination of a binary cross-entropy loss and a Dice loss [21], which helped handle the imbalance of the input data in our problem.

### 2.5. Statistical Analysis

To investigate the effect of training set noise, we used each of the list-mode sets of reconstructions for *Set1* to train lesion-detection models using 5 separate runs with different random seeds. Then, we evaluated their performance on the testing set of Network2. For the effect of training sample size, we trained lesion detection models based on 25%, 50%, 75% and 100% of each acquisition in *Set1*. For the 5 runs on each training dataset, we used the same *Set1* validation set and *Set2* test set for validation and out-of-domain evaluation, respectively. In the testing stage, we directly applied each Network1-trained model on the *Set2* testing set to produce a prediction map for each input image, and used a threshold (i.e., 0) to binarize the map to identify lesions. Then we applied a noise filter of 20 pixels and excluded predictions below that threshold. We used positive predictive value, sensitivity and *F_1_* score as model evaluation metrics [11,12]. With a connected component analysis, we used the Hungarian algorithm [22] to match gold-standard annotations with automated predictions to handle potentially multiple lesions per subject. A detected lesion was considered true positive (*TP*) if the intersection over union (IoU) between this lesion and a gold-standard lesion annotation was greater than zero [11,12]; otherwise, the detected lesion was viewed as false positive (*FP*). Any gold-standard lesion with no matched detection is defined as false negative (*FN*). With these definitions, we calculated positive predictive value *PPV* = *TP*/(*TP* + *FP*), sensitivity = *TP*/(*TP* + *FN*) and *F*_1_ score *F*_1_ = (2 × *PPV* × *sensitivity*) */* (*PPV* + *sensitivity*), for the test set.

## 3. Results

The patient demographics for *Set1* are shown in Table 1, and those for the *Set2* have been previously reported [12]. When evaluating *Network1*-trained DL models with the *Set2* testing dataset (Table 2 and Figure 1), the *F*_1_ score progressively improved with decreasing acquisition time, i.e., higher noise level images showed improved performance, with *F*_1_ score increasing from 0.657 at 5-min acquisition to 0.713 at 2-min acquisition. This demonstrates an improved *F*_1_ score approaching the upper-bound limit model (*F*_1_ = 0.755) of performance. This upper bound limit, indicating best possible performance, uses training data from *Set2* and tests the model with the in-domain *Set2* testing dataset. Similarly, PPV increased with decreasing acquisition time, while sensitivity only slightly decreased. The *F*_1_ score of the DL model trained with the original clinical reconstruction from *Set1* (Q.Clear; *F*_1_ = 0.614) was significantly worse compared to the model trained with *Set1′s* 2-min reconstructed dataset (VPFXS 2 min; *F*_1_ = 0.713; *p*-value = 0.006). In addition, as the *F*_1_ score of the noisier images was better on the out-of-domain test data, the highest noise 2-min VPFXS images showed performance comparable to and not significantly different from the upper-bound model (*F*_1_ = 0.713 vs. 0.755; *p*-value = 0.103). Similarly, there was no significant difference between the upper-bound model and the model trained with the VPFXS 3 min (*p*-value = 0.087); whereas the *F*_1_ score of the VPFXS 4 min model, that of the VPFXS 5 min model and that of the Q.Clear model were all significantly smaller than that of the upper-bound model (*p*-value = 0.013, 0.026, 0.006, respectively).

The results of the DL models with different training sample sizes are shown in Table 3. The *F*_1_ score progressively increased from 0.478 (25% training set) to 0.713 (100% training set). The *F*1 score of the model trained with the 100% VPFXS 2-min dataset was significantly higher than that of the one trained with the 25% VPFXS 2-min dataset (*p*-value ≤ 0.001). Additionally, the *F*_1_ score increased as the sample size increased, with *p*-value = 0.012 between 25% and 50%, *p*-value = 0.046 between 50% and 75% and *p*-value = 0.049 between 75% and 100%. 

Several examples are shown in Figure 2. In the qualitative prediction results of the VPFXS 2-min model (Figure 2), the “True Positive” columns from Subjects A & B are examples in which the prediction (Row 2) agrees with the corresponding gold standard (Row 3). Similarly, the “False Negative” and “False Positive” examples in Subjects B & C are also presented.

## 4. Discussion

This study demonstrates the feasibility of addressing domain shift by better noise matching. In this study, the out-of-domain PET images from *Set2* are much noisier compared to those from the original *Set1*. The level of domain shift between the original clinical *Set1* Q.Clear dataset (*F*_1_ = 0.614) and *Set2* (*F*_1_ = 0.755) was significant (*p*-value = 0.006). With list-mode reconstruction, we generated PET images with higher levels of noise by choosing shorter acquisition times. List-mode reconstructions of *Set1* with different shorter acquisition times demonstrated significantly improved *F*_1_ score on unseen reconstructions from *Set2* from 5 min (0.614) to 2 min (0.713). Finally, the performance of this 2-min acquisition (*F*_1_ = 0.713) improved to a level that was not significantly different compared to the performance of the upper-bound limit (*Set2*, *F*_1_ = 0.755; *p*-value = 0.103). This demonstrates the potential of eliminating the domain shift by better matching noise properties.

This deep lesion detection network for the ^68^Ga-DOTATATE PET dataset requires a relatively large dataset to achieve better performance. In the experiments of training networks with different percentages of the full training set, there is consecutively a significant increase in *F*_1_ score as the training sample size increases, with *F*_1_ from 0.478 to 0.616 (*p*-value = 0.012) between 25% and 50%, *F*_1_ from 0.616 to 0.662 (*p*-value = 0.046) between 50% and 75% and *F*_1_ from 0.662 to 0.713 (*p*-value = 0.049) between 75% and 100%. The improvement with larger dataset sizes emphasizes the potential for larger training datasets to improve DL algorithm performance. Although an even larger dataset could further improve the performance in our study, we attained a high level not significantly different compared to the upper bound limit.

Another minor difference between standard clinical reconstructions and our list-mode reconstructions was in the choice of reconstruction algorithm. A noisier reconstruction (VPFXS) was chosen compared to the more contemporary and lower noise reconstruction (Q.Clear) used in clinical scanning protocols. This demonstrates the ability to further alter the noise properties by using specific reconstruction techniques. This feasibility study did not optimize other reconstruction parameters (post-reconstruction filtering, depth dependent resolution recovery, time-of-flight or other factors.) which could be tested to further improve the matching of properties in the different datasets.

To our knowledge, this is the first report of using list-mode reconstructions to better match training datasets to out of domain target datasets, which subsequently demonstrate improved DL performance. Although this report is novel with respect to the use of list-mode data to improve PET lesion detection, other reports have shown that image pre-processing can significantly improve convolutional neural network performance in FDG PET lesion segmentation [23] and MRI lesion characterization [24].

The ability to retrospectively reconstruct PET list-mode data with different imaging characteristics may allow a much broader diversification of PET imaging properties in the training data. Specific imaging properties could be applied to reconstruct PET images with varying levels of image noise, image spatial resolution and specific corrections such as scatter correction, time-of-flight (TOF) corrections, depth dependent resolution recovery corrections and post- reconstruction filtering. Matching these properties to a target dataset may potentially further improve PET lesion detection performance.

This work complements our prior work, which demonstrated significantly improved PET lesion detection performance with advanced DL techniques including domain adaptation, single domain generalization and 3-dimensional lesion detection [25,26,27]. These techniques were also combined with previously established techniques such as region-guided generative adversarial network (RG-GAN) for lesion-preserved image-to-image translation and data augmentation [27]. Taken together, both our data preprocessing techniques and our advanced DL techniques show great potential for improvements in DL performance when testing datasets from outside institutions with differing PET instrumentation, acquisition and processing protocols.

## 5. Conclusions

List-mode data from modern PET scanners can be reconstructed to better match the higher noise properties of reconstructions from an older-generation PET scanner. Reuse of the existing data and their associated annotations can dramatically reduce the cost and effort to generate these better matching datasets. These reconstructed datasets can significantly improve the performance of existing DL algorithms and, thus, provide an efficient, low-cost method to rapidly improve DL algorithm generalizability.

## Figures and Tables

**Figure 1 bioengineering-11-00226-f001:**
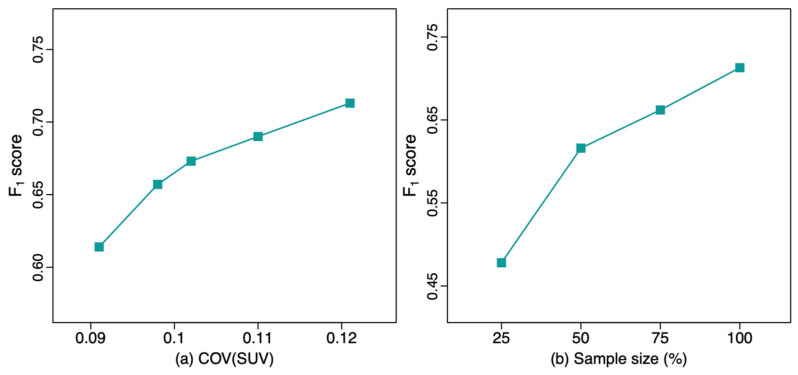
The *F*_1_ score of the lesion detection model with different values of (**a**) COV(SUV) and (**b**) sample size.

**Figure 2 bioengineering-11-00226-f002:**
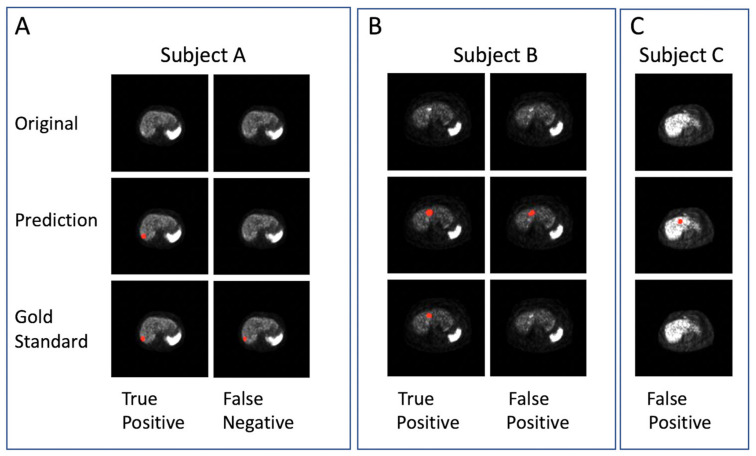
Examples of DL lesion detection in transaxial ^68^Ga DOTATATE PET. Lesion predictions and gold-standard annotations are marked in red. (Top row) Original images, (middle row) DL prediction, and (bottom row) Gold standard. (**A**–**C**) Three different patient examples: (**A**) true positive and false negative, (**B**) true positive and false positive and (**C**) false positive.

**Table 1 bioengineering-11-00226-t001:** Patient demographics and baseline characteristics. Value for mean age is mean (standard deviation). Values for other parameters are number (percentage).

Parameter	Value
Mean age (years)	61.4 (14.09)
Women	61.4
Men	61.2
Sex (no. of patients)	
Women	40 (48%)
Men	43 (52%)
Tumor present in liver	
Yes	41 (49%)
No	42 (51%)
Primary tumor site	
Small bowel	32 (38%)
Pancreas	25 (30%)
Stomach	5 (6.5%)
Lung	5 (6.5%)
Head and neck	5 (6.5%)
Large bowel	2 (2%)
Adrenal	3 (3%)
None (normal scan)	6 (7.5%)
Ki-67 index	
Low/intermediate grade (≤20%)	51 (62%)
High grade (>20%)	1 (1%)
No pathology report	31 (37%)

**Table 2 bioengineering-11-00226-t002:** Lesion detection evaluation on the unseen *Set2*: effect of noise levels. Each method was run 5 times, and the mean and standard deviation (SD) of each metric are reported: mean (SD). We also present the noise level of each dataset in terms of COV of SUV: mean (SD). “*” means significant difference compared with *Scanner2 F*_1_ score.

Training Set	COV	*F* _1_	PPV	Sensitivity
*Set1* Q.Clear	0.091 (0.027)	0.614 * (0.052)	0.706 (0.119)	0.565 (0.111)
*Set1* VPFXS 5 min	0.098 (0.027)	0.657 * (0.033)	0.637 (0.105)	0.695 (0.059)
*Set1* VPFXS 4 min	0.102 (0.027)	0.673 * (0.027)	0.663 (0.087)	0.694 (0.048)
*Set1* VPFXS 3 min	0.110 (0.029)	0.690 (0.034)	0.707 (0.087)	0.681 (0.025)
*Set1* VPFXS 2 min	0.121 (0.030)	0.713 (0.028)	0.758 (0.087)	0.680 (0.039)
*Set2*	0.198 (0.040)	0.755 (0.043)	0.817 (0.036)	0.706 (0.070)

**Table 3 bioengineering-11-00226-t003:** Lesion detection evaluation on the unseen test dataset: effect of training sample size. Each method was run 5 times, and the mean and standard deviation (SD) of each metric are reported: mean (SD). “*” means significant difference compared with 100% *Scanner1* F**_1_** score.

Training Sample Size	F_1_	PPV	Sensitivity
25% *Set1* VPFXS 2 min	0.478 * (0.044)	0.620 (0.049)	0.392 (0.055)
50% *Set1* VPFXS 2 min	0.616 * (0.046)	0.882 (0.028)	0.475 (0.054)
75% *Set1* VPFXS 2 min	0.662 * (0.019)	0.745 (0.051)	0.598 (0.031)
100% *Set1* VPFXS 2 min	0.713 (0.028)	0.758 (0.087)	0.680 (0.039)

## Data Availability

The datasets presented in this article are not readily available because they are under the auspices of the institutional data privacy statement. Requests to access the datasets should be directed to the corresponding author.

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
