# Peer review of "Improving Generalizability of PET DL Algorithms: List-Mode Reconstructions Improve DOTATATE PET Hepatic Lesion Detection Performance"

_bioengineering, 2024, doi:10.3390/bioengineering11030226_

Round 1

Reviewer 1 Report

Comments and Suggestions for Authors

This paper presents a study of the feasibility of improving DL algorithm performance by better matching the background noise in training datasets to the noise level in testing datasets coming from another source (out-of-domain datasets). This is accomplished by generating a set of reconstructed images with different acquisition times based on the same PET list-mode data, such that only the noise level is different between the datasets. In addition, the effect of training sample size on the cross-domain performance is also investigated. The results are convincing and relevant for continued development and implementation of AI image analysis tools to work across various institution with different imaging equipment and protocols.

The manuscript is well written, the methodology and results are clearly presented, and relevant references to previous work is included.

The work is recommended for publication in the present form.

Specific comments:

Figure 1: The images appear very dark, making it very difficult to see the relevant features on the “original” images. Consider adjusting (increasing) brightness and contrast for better appearance.

Supplemental figure S1: Input image as well as the prediction maps appear very dark. brightness and contrast should be adjusted.

Reviewer 2 Report

Comments and Suggestions for Authors

I examined your article titled "Improving Generalizability of PET DL Algorithms: List-Mode Reconstructions Improve DOTATATE PET Hepatic Lesion Detection Performance" in detail. I accept that the work you have done is valuable, but I would like to point out that there are major shortcomings in the study. The deficiencies in the study are presented below.

-Abstract should be simplified. In particular, how the data set is divided and repeating sentences should be avoided. In the abstract, the importance of two data sets is highlighted. This is an important stage, of course, but why the study was done, what its purpose was and the proposed model should also be brought to the fore.

-A paragraph should be added at the end of the Introduction section about the innovations of the article and its contributions to the literature.

-A paragraph regarding the organization of the article should be added to the last paragraph of the Introduction section.

- Under the Image Acquisition and Datasets heading, there should be the number of patients and images in each data set and sample images from the data set. Because more data sets are highlighted in the article. When testing the model, was patient level or image level applied?

-I think the application results were obtained using u-net architecture. No information is given about this part in the article. In addition, general information is given under the subheadings of the material and method. There is no information about how it was used in the study.

-I think that especially the result section should be reorganized and supported with figures in this section. The first paragraph of the Result section completely drowns the reader in numerical data. For example, this section should be expanded with graphics and confusion matrix-like structures.

-I would like to point out that although I examined the article in great detail, I could not really capture the essence of the article. As I understand it, the importance of two data sets has been emphasized. Please make any additions that will prove me wrong on this issue.

-I believe that the quality of the article will increase if the deficiencies I have mentioned are corrected. I wish conveniences.  Best regards.

Comments on the Quality of English Language

It is important to review spelling and grammatical errors.

Reviewer 3 Report

Comments and Suggestions for Authors

The paper requires major revisions.

1. Provide more details about the specific deep learning architecture used in your study, such as the number of layers, type of activation functions, and any unique modifications to the U-Net architecture.

2. How did you handle class imbalance in your dataset, especially given the rarity of gastroenteropancreatic neuroendocrine tumors (GEP-NETs)?

3. Regarding the list-mode reconstructions, could you elaborate on the specific variations applied to better match the noise properties between training and out-of-domain testing datasets?

4. In the statistical analysis section, you mentioned using the Hungarian algorithm for matching gold-standard annotations with automated predictions. Can you explain why you chose this specific algorithm and how it contributes to your evaluation?

5. What criteria were used to define normal liver background in the semiautomated MIM workflow, and how robust is this method across different datasets?

6. When discussing the effect of training sample size, did you observe any trends or patterns in terms of the network's performance with a smaller training set?

7. How did you determine the optimal threshold for binarizing the prediction maps during the testing stage, and did you explore different threshold values?

8. Could you provide more insights into the decision to use a linear combination of binary cross-entropy loss and Dice loss in optimizing the network? How did this combination contribute to handling data imbalance?

9. In the Materials and Methods section, you mentioned random seed runs for each list-mode set of reconstructions. How did the variability in random seeds affect your results, and were there any considerations or precautions taken to address this variability?

10. What are the potential limitations of using the modified U-Net architecture in your study, and how might these limitations impact the generalizability of your findings?

11. In the Results section, you discussed the progressively improving F1 score with decreasing acquisition time. Were there any unexpected findings or challenges encountered during this analysis?

12. Can you elaborate on the significance of the COV (coefficient of variation) of SUV measurements in quantifying image noise and how it relates to the overall performance of the deep learning models? Add suitable references to the site with [PMID: 37701174, PMID: 37993801 ]

13. How did you ensure the quality and accuracy of the lesion contours generated using the MIM workflow, especially when transferring contours from high-quality Q.Clear reconstructions to VPFXS reconstructions?

14. Considering the potential impact of spatial resolution, image noise, and image processing differences between datasets, how did you address these factors in the training process to improve model generalizability?

15. Were there any specific challenges or limitations encountered when working with the older-generation analog PET/CT scanner (Scanner2) in terms of image quality and dataset variability?

16. What considerations were taken into account when selecting the reconstruction parameters for both Set1 and Set2, and how did these choices impact the overall results?

17. In the discussion of patient demographics, you provided information on tumor presence, primary tumor site, and Ki-67 index. How did these factors influence the training and evaluation of the deep learning models?

18. Could you discuss potential future directions or extensions of your study, especially in terms of addressing challenges related to the scarcity of well-annotated data for GEP-NETs?

19. How do you envision the practical application of your findings in the clinical setting, and what steps could be taken to facilitate the adoption of your proposed approach in real-world scenarios?

Round 2

Reviewer 3 Report

Comments and Suggestions for Authors

After a thorough evaluation of the revised manuscript, I am pleased to report that the authors have successfully incorporated all of the recommended changes, which have significantly improved the quality of the paper.